# Mesenchymal Stem Cell-Mediated Deep Tumor Delivery of Gold Nanorod for Photothermal Therapy

**DOI:** 10.3390/nano12193410

**Published:** 2022-09-28

**Authors:** Wan Su Yun, Man Kyu Shim, Seungho Lim, Sukyung Song, Jinseong Kim, Suah Yang, Hee Sook Hwang, Mi Ra Kim, Hong Yeol Yoon, Dong-Kwon Lim, In-Cheol Sun, Kwangmeyung Kim

**Affiliations:** 1KU-KIST Graduate School of Converging Science and Technology, Korea University, Seoul 02841, Korea; 2Biomedical Research Institute, Korea Institute of Science and Technology (KIST), Seoul 02792, Korea; 3Department of Pharmaceutical Engineering, Dankook University, Cheonan 31116, Korea; 4Department of Otorhinolaryngology-Head and Neck Surgery, Haeundae Paik Hospital, Inje University College of Medicine, Busan 48108, Korea; 5College of Pharmacy, Graduate School of Pharmaceutical Sciences, Ewha Womans University, Seoul 03760, Korea

**Keywords:** mesenchymal stem cell, gold nanorod, drug delivery, deep tumor penetration, photothermal therapy

## Abstract

Gold nanoparticles (AuNPs) with various sizes and morphologies have been extensively investigated for effective photothermal therapy (PTT) against multiple cancer types. However, a highly dynamic and complex tumor microenvironment (TME) considerably reduces the efficacy of PTT by limiting deep tumor penetration of AuNPs. Herein, we propose a mesenchymal stem cell (MSC)-mediated deep tumor delivery of gold nanorod (AuNR) for a potent PTT. First, MSCs are treated with tetraacylated N-azidomannosamine (Ac_4_ManNAz) to introduce modifiable azide (N_3_) groups on the cell surface via metabolic glycoengineering. Then, AuNRs modified with bio-orthogonal click molecules of bicyclo[6.1.0]nonyne (AuNR@BCN) are chemically conjugated to the N_3_ groups on the MSC surface by copper-free click chemistry reaction, resulting in AuNR@MSCs. In cultured MSCs, the appropriate condition to incorporate the AuNR into the MSCs is optimized; in addition, the photothermal efficiency of AuNR-MSCs under light irradiation are assessed, showing efficient heat generation in vitro. In colon tumor-bearing mice, intravenously injected AuNR@MSCs efficiently accumulate within the tumor tissues by allowing deep tissue penetration owing to the tumor homing effect by natural tumor tropism of AuNR@MSCs. Upon localized light irradiation, the AuNR@MSCs significantly inhibit colon tumor growth by the enhanced photothermal effect compared to conventional AuNRs. Collectively, this study shows a promising approach of MSCs-mediated deep tumor delivery of AuNR for effective PTT.

## 1. Introduction

Light-mediated therapeutic approaches have emerged as an effective cancer treatment owing to unique advantages, such as minimal invasiveness, high selectivity, and spatiotemporal control [1,2,3]. Photothermal therapy (PTT), which can convert light to heat via photothermal agents, causes irreversible damage in cancer cells by inducing localized heat depending on the magnitude of the light exposure time and its fluence in a noninvasive manner [4]. Importantly, gold nanoparticles (AuNPs) that have various size and morphology of spheres, rods, shells, clusters, and cages have been extensively investigated for effective PTT in diverse cancer types [5,6,7]. Importantly, the AuNPs efficiently accumulate within the tumor tissues owing to the enhanced permeability and retention (EPR) effect based on their nano-sized structure; in addition, the tumor targeting efficiency can be further enhanced by modification with active-targeting ligands because AuNPs have great amenability for surface functionalization [8,9,10,11]. However, tumor tissues have a highly dynamic and complex microenvironment (TME), which is characterized by (i) regional blood flow to the tumors and increased vessel permeability; (ii) high stiffness, interstitial fluid pressure (IFP), and solid stress; and (iii) physical barriers imposed by the extracellular matrix (ECM) and increased ECM cross-linking [12,13]. These factors considerably reduce the tumor targeting efficiency of passive and active targeting strategies by limiting the deep tumor penetration of nanoparticles [14]. Therefore, there is a desperate need for a promising strategy to deliver AuNR to the deep tumor tissues for improving PTT efficiency.

Mesenchymal stem cells (MSCs) that can be easily isolated from fat liver, muscles, bone marrow, and many other places has become an attractive delivery system as a biovesicles to transport nanoparticles to tumor tissues [15]. The MSCs, that often lack major histocompatibility complex-II (MHC class II), prevent T cell responses and thus evade the immune systems complications in human and animal models [16]. Notably, MSCs exhibit natural tumor tropism premised on the site-specific expression of growth factors, such as epidermal growth factor, platelet-derived growth factor, and stromal-derived factor-1 [17]. Accordingly, they have an intrinsic homing nature to tumors by the activation of tumor-associated chemokine receptors; in addition to their tumor-targeting ability, recent studies have shown that MSCs hold a homing effect against specific tumors, which can be optimized by employing the appropriate types of MSCs to target a specific type of tumor [18]. Based on these distinct advantages, MSCs have been widely studied to enhance the deep tumor penetration of nanoparticles, which is mainly achieved by intracellular loading methods [19]. This is because the MSCs should encapsulate the nanoparticles based on physical and non-specific loading approaches owing to a lack of specific receptors compared to cancer cells [20]. However, this approach can diminish the natural functionality of MSCs and alter their in vivo fate; most importantly, it also shows limited drug loading efficiency, resulting in poor therapeutic efficacy.

Herein, we propose an MSCs-mediated deep tumor delivery of gold nanorod (AuNR) for a potent PTT by the incorporation of AuNR into the MSCs through metabolic glycoengineering and copper-free click chemistry reaction. The metabolic glycoengineering can incorporate modifiable chemical groups into the surface glycans of the cells [21,22,23]. When the cells are treated with unnatural metabolites such as Ac_4_ManNAz, Ac_4_GalNAz, and Ac_4_GlcNAz, they utilize them to build blocks via an intrinsic glycan mechanism; as a result, azide (N_3_) groups in such metabolites are exogenously generated on the cell surface [24]. The successful introduction of N_3_ groups on the cell surface using unnatural metabolites was evaluated in stem cells as well as in various cancer (colon, breast, glioma, and prostate cancers) cells and normal (human fibroblast, cardiomyocytes, and human umbilical vein endothelial) cells, which was extensively employed for the biomedical application of tumor-specific imaging and drug delivery [21,23,25]. In particular, nanoparticles containing bio-orthogonal click molecules of bicyclo[6.1.0]nonyne (BCN) or dibenzylcyclooctyne (DBCO) can be chemically conjugated with N_3_ group in the unnatural glycans by copper-free click chemistry reaction [25]. The important benefits of this strategy are in allowing high-amount loading of nanoparticles without affecting the intrinsic functions and fates of MSCs. Previous study has demonstrated a successful in vivo tracking of MSCs via magnetic resonance (MR) imaging by labeling the cells with superparamagnetic iron oxide nanoparticles via metabolic glycoengineering [26]. In this study, AuNRs modified with bio-orthogonal click molecules of BCN (AuNR@BCN) are prepared, and modifiable N_3_ groups are introduced in MSCs by treating unnatural metabolites (Ac_4_ManNAz) for metabolic glycoengineering; eventually, AuNR@BCN is incorporated into the MSCs (AuNR@MSCs) via copper-free click chemistry reaction (Figure 1a). The AuNRs that are incorporated into the MSCs via copper-free click chemistry reaction of BCN and N_3_ on the cell surface efficiently internalized into the cytoplasm of the MSCs owing to a membrane turnover mechanism. The biocompatibility of these nanoparticle internalization mechanisms by metabolic glycoengineering and copper-free click chemistry reaction has been evaluated, and it did not affect stem cell function [25]. When the AuNR@MSCs are intravenously injected into the colon tumor models, they efficiently penetrate deep inside the tumor tissues via a stem cell homing effect and induce local hyperthermia upon light irradiation (Figure 1b). In cell culture systems, the appropriate condition to incorporate the AuNR into the MSCs is optimized. In addition, the photothermal efficiency of AuNR-MSCs are assessed under light irradiation, showing efficient heat generation owing to a high AuNR-loading efficiency compared to conventional intracellular loading methods. The superior antitumor efficacy of AuNR-MSCs by deep tumor penetration is assessed in colon tumor-bearing mice. Collectively, this study shows a promising approach of MSCs-mediated deep tumor delivery of AuNR for an effective PTT.

### 1.1. Reagents

Gold(III) chloride trihydrate, silver nitrate, hydrochloric acid, tetraethyl orthosilicate (TEOS), (3-aminopropyl)trimethoxysilane (APTMS), bicyclo[6.1.0]non-4-yn-9-ylmethyl N-succinimidyl carbonate (BCN-NHS), L-ascorbic acid, and sodium borohydride were purchased from Sigma Aldrich (Oakville, ON, USA). Cetyltrimethylammonium bromide (CTAB) was purchased from Tokyo Chemical Industry (TCI, Tokyo, Japan). Cyanine5.5 NHS ester was purchased from Lumiprobe (Hunt Valley, MD, USA). Tem grid (Carbon Film 200 Mesh copper) was purchased from Electron Microscopy Sciences (Hatfield, PA, USA). Human adipose-derived mesenchymal stem cells (MSCs) and HT29 (human colon adenocarcinoma) were purchased from American Type Culture Collection (ATCC; Manassas, VA, USA). RPMI 1640 medium, fetal bovine serum (FBS), penicillin, and streptomycin were purchased from WELGENE Inc. (Daegu, Korea). Minimum Essential Medium α, fetal bovine serum (FBS), Dulbecco’s phosphate-buffered saline (DPBS), EZ-Link phosphine-PEG3-biotin, and streptavidin-conjugated horseradish peroxidase (streptavidin-HRP) were purchased from Thermo Fisher Scientific (Waltham, MA, USA). Cell counting kit-8 (CCK-8) was purchased from Vitascientific (Beltsville, MD, USA). Tetraacetylated N-azidoacetyl mannosamine (Ac_4_ManNAz) was purchased from Invitrogen (Rockford, IL, USA).

### 1.2. Preparation and Characterization of AuNR@BCN

To incorporate the AuNRs into the MSCs via copper-free click chemistry reaction, BCN groups were introduced in the AuNRs. First, CTAB (3.6445 g, 0.1 M), HAuCl_4_ (19.6865 mg, 0.01 M), AgNO_3_ (1.69 mg), and ascorbic acid (176 mg, 0.1 M) were stirred at 1100 rpm for 3 min, resulting in AuNR@CTAB. Then, AuNR@CTAB was mixed with 0.8 mM CTAB solution, followed by incubation with 0.1 M NaOH and 1 M TEOS (20% *v/v* in MeOH) for 15 min. The resulting AuNR@SiO_2_ was further mixed with APTMS (0.04 mL) at 4 °C for 24 h to yield AuNR@NH_2_. Finally, the AuNR@NH_2_, BCN-NHS ester, and Cy5.5-NHS ester were dissolved in DMSO, and the solutions were stirred for 24 h at 4 °C, resulting in AuNR@BCN.

### 1.3. Preparation and Characterization of AuNR@MSCs

In order to generate azide (N_3_) groups on the MSC surface via metabolic glycoengineering, 5 × 10^5^ cells were seeded in the cell culture dishes, followed by treatment with 20 μM Ac_4_ManNAz for 48 h. Then, the cells were further incubated with 10 mM BCN-Cy5.5 at 37 °C for 2 h. For the fluorescence imaging, the cells were washed twice with DPBS, fixed with 4% paraformaldehyde for 20 min, and stained with 4′,6-diamidino-2-phenylindole (DAPI) for 15 min. The N_3_ generation in the MSCs was observed via a Leica TCS SP8 laser-scanning confocal microscope (Leica Microsystems GmbH; Wetzlar, Germany) equipped with diode (405 nm), Ar (458, 488, 514 nm), and He-Ne (633 nm) lasers. To directly visualize the N_3_ generation, fluorescence dye Cy5.5-conjugated BCN (BCN-Cy5.5, 2 μM), which can be chemically conjugated with N_3_ on the cell surface, was further incubated with Ac_4_ManNAz-treated MSCs. The successful N_3_ generation on the MSC surface was also evaluated via Western blot analysis. Briefly, the Ac_4_ManNAz-treated MSCs were lysed using RIPA buffer (1% SDS, 100 mM Tris-HCl, pH 7.4) with protease inhibitor for 1 h at 4 °C. After protein quantification by bicinchoninic acid protein assay (BCA), each 5 mg/mL cell lysate was mixed with 5 mM phophine-PEG_3_-biotin and incubated at room temperature for 12 h. The proteins from each sample were mixed with 1× sodium dodecyl sulfate (SDS) gel-loading dye and boiled for 5 min. Then, 5 μg of proteins was separated by 12% SDS-polyacrylamide gel electrophoresis and subsequently transferred onto polyvinylidene fluoride (PVDF). The membranes were incubated with 1× TBST (10 mol/L Tris, 100 mol/L NaCl, and 0.1% Tween 20, pH 7.4) containing 5% bovine serum albumin (BSA). Finally, the membranes were further incubated with 1× TBST containing streptavidin-HRP for 2 h at room temperature, and the protein band was visualized by an enhanced chemiluminescence kit.

To incorporate the AuNR@BCN in the Ac_4_ManNAz-treated MSCs via copper-free click chemistry reaction, 5 × 10^5^ Ac_4_ManNAz-treated MSCs were incubated with AuNR@BCN at a 200 μg/mL concentration for 6 h. Successful AuNR incorporation in the MSCs (AuNR@MSCs) was observed via a Leica TCS SP8 laser-scanning confocal microscope (Leica Microsystems GmbH; Wetzlar, Germany) equipped with diode (405 nm), Ar (458, 488, 514 nm), and He-Ne (633 nm) lasers. The photothermal efficiency of AuNR@MSCs was assessed after light irradiation with a power of 1.0 W/cm^2^ (CW laser, Changchun New Industries Optoelectronics Tech. Co. Ltd., Changchun, China). The real-time temperature and thermal images were recorded using a digital thermometer (HH506A, OMEGA, Norwalk, CT 06854, USA) and IR camera (E6, FLIR Systems, Seoul, Korea), respectively.

### 1.4. Deep Tumor Penetration of AuNR@MSCs in Colon Tumor Models

The deep tumor penetration of AuNR@MSCs was assessed in colon tumor-bearing mice, which were prepared via subcutaneous inoculation of 1 × 10^7^ HT29 cells into the left flank. When the tumor volumes were approximately 200 mm^3^, AuNR@MSCs or AuNR@BCN with an equivalent concentration of 5 mg/kg of AuNR were intravenously injected into the mice. To administrate AuNR@MSCs with 5 mg/kg AuNR concentration, the amount of AuNR@MSCs were quantified by comparing with Cy5.5 fluorescence intensities of 5 mg/kg of AuNR@BCN (modified with Cy5.5). The tumor accumulation was observed by noninvasive near-infrared fluorescence (NIRF) imaging via an IVIS Lumina Series III system (PerkinElmer; Waltham, MA, USA). On day 5 after treatment, the major organs and tumor tissues were collected from the mice for ex vivo NIRF imaging, and fluorescence intensities were quantified using Living Image software (PerkinElmer, Waltham, MA, USA). In addition, tumor tissues from mice were cut into 10-μm thick sections for histology. Slide-mounted tumor sections were washed with DPBS three times and stained with GFP fluorescent dye-conjugated CD31 antibody at 4 °C for 12 h. Then, the nuclei of the tumor tissues were stained with DAPI for 15 min at dark condition and analyzed using a Leica TCS SP8 confocal laser scanning microscope (Fasanenstrasse 71, 10719 Berlin, Germany).

### 1.5. Therapeutic Efficacy and Toxicity Evaluation in Colon Tumor Models

The therapeutic efficacy and toxicity of AuNR@MSCs were assessed in HT29 tumor-bearing mice. Briefly, the mice were randomly divided into four groups: (i) saline, (ii) light irradiation only (Laser only), (iii) AuNR@BCN with light irradiation (AuNR+L), and (iv) AuNR@MSCs with light irradiation (AuNR@MSC+L). When the tumor volumes were approximately 100 mm^3^, AuNR@MSCs or AuNR@BCN with an equivalent concentration of 5 mg/kg of AuNR were intravenously injected into the HT29 tumor-bearing mice. In addition, tumor tissues in the Laser only, AuNR+L, and AuNR@MSC+L groups were locally irradiated by light with a power of 1.0 W/cm^2^ for 5 min; light irradiation was performed after 3 days of AuNR@MSCs or AuNR@BCN treatment. The therapeutic efficacy was assessed by measuring the tumor volumes, calculated as the largest diameter × smallest diameter^2^ × 0.53. The tumor volumes and body weights were measured every day, and mice with a tumor size of 2000 mm^3^ or higher were counted as dead.

### 1.6. Statistics

Statistical analyses were performed using GraphPad Prism 9 software (San Diego, CA 92108, USA). The statistical significance between two groups was analyzed using Student’s *t*-test. One-way analysis of variance (ANOVA) was performed for comparisons of more than two groups, and multiple comparisons were analyzed using the Tukey–Kramer *post-hoc* test. Survival data were plotted as Kaplan–Meier curves and analyzed using the log-rank test. In the figures, statistical significance is indicated with asterisks (* *p* < 0.05, ** *p* < 0.01, *** *p* < 0.001).

### 1.7. Data Availability

All relevant data are available with the article and its Appendix A, or are available from the corresponding authors upon reasonable request.

## 2. Results and Discussion

### 2.1. Preparation and Characterization of AuNR@BCN

To incorporate the AuNR into the human adipose tissue-derived mesenchymal stem cells (MSCs) via metabolic glycoengineering and copper-free click chemistry reaction, AuNRs were modified with BCN (AuNR@BCN; Appendix A). The cetyltrimethylammonium bromide-stabilized gold nanorods (AuNR@CTAB) were used as a platform AuNPs due to their great amenability to modify its size, shape, and surface [5]. First, AuNR@CTAB was coated with silica (AuNR@SiO_2_) because the protecting and shielding of the AuNR surface with silica shells can increase their stability and biocompatibility, resulting in the enhanced effectiveness of PTT [27]. The transmission electron microscope (TEM) images of the AuNR@SiO_2_ clearly showed the silica-coated areas on the AuNR surface with a thickness of ~20 nm after modification of AuNR@CTAB (Figure 1a). To incorporate the bio-orthogonal click molecules onto the surface of AuNR, the AuNR@SiO_2_ was functionalized with 3-aminopropyltriethoxysilane (APTES) using the Stöber method, resulting in AuNR@NH_2_ [28]; finally, AuNR@BCN was obtained by the chemical conjugation of AuNR@NH_2_ with BCN-succinimidyl carbonate (BCN-NHS ester) via copper-free click chemistry reaction. As shown in Figure 1a, there were no significant morphological changes in the AuNR@NH_2_ and AuNR@BCN compared to AuNR@SiO_2_, as confirmed by TEM images. The average size of AuNR@CTAB (about 80–90 nm) was slightly increased after silica-coating, wherein the AuNR@SiO_2_, AuNR@NH_2_, and AuNR@BCN have a similar average size of approximately 130–140 nm (Figure 1b). The zeta potential of AuNR@CTAB with high positive charge was changed to neutral after surface silica-coating, but that was reversed as a positive charge after modification with BCN (Figure 1c). Compared with the AuNR@CTAB (800 nm), the longitudinal SPR peaks of AuNR@SiO_2_, AuNR@NH_2_, and AuNR@BCN moved to 820 nm owing to the altered local environment around the AuNRs after silica-shell coating (Figure 1d). Next, the biocompatibility of AuNR@BCN was evaluated in the MSCs, showing no significant cytotoxicity after 48 h of treatment, with concentrations from 0 to 200 μg/mL (Figure 1e). Finally, the photothermal efficiency of each AuNR was assessed under light irradiation with a power of 1.0 W/cm^2^. As shown in the photothermal images, the local temperature in the tubes was significantly increased up to 56 °C by AuNR@CTAB; in addition, AuNR@SiO_2_, AuNR@NH_2_, and AuNR@BCN showed comparable photothermal efficiency with AuNR@CTAB in the same experimental condition (Figure 1f,g). These results indicate that surface modification to incorporate the bio-orthogonal click molecules onto the surface of AuNR did not influence their basal photothermal efficiency. Taken together, as a photothermal agent modified with bio-orthogonal click molecules for incorporation into the MSCs, AuNR@BCN was successfully prepared without affecting the intrinsic characteristics of AuNRs, such as morphology, photothermal efficiency, and biocompatibility.

### 2.2. Optimization for Generation of Azide Groups on the Stem Cell Surface

In order to generate azide (N_3_) groups on the surface of MSCs without affecting their intrinsic functions and fates, appropriate treatment periods and times of unnatural metabolites (Ac_4_ManNAz) were optimized in vitro. When the MSCs were incubated with different concentrations of Ac_4_ManNAz (0–50 μM) for 48 h, the amount of N_3_ generated on the cell surface was gradually increased in a dose-dependent manner up to 20 μM, but that was similar in the MSCs treated with 20 or 50 μM (Figure 2a). The N_3_ generation on the MSC surface was further visualized via cellular fluorescence imaging, wherein the MSCs were further incubated with BCN modified with fluorescent dye, Cy5.5 (BCN-Cy5.5), for 2 h after treatment and with Ac_4_ManNAz (0–50 μM) for 48 h (Figure 2b). As expected, a strong Cy5.5 fluorescence signals on the cell surface were clearly observed in the MSCs treated with 20 μM Ac_4_ManNAz, and those were similar with 50 μM Ac_4_ManNAz-treated MSCs. Next, the cytotoxicity was evaluated by the different concentrations of Ac_4_ManNAz treatments in MSCs. The result showed that treatment of up to 20 μM Ac_4_ManNAz did not induce significant cell death of MSCs, but the cell viability of MSCs was significantly decreased compared to naive cells after 48 h of 50 μM Ac_4_ManNAz treatment (Figure 2c). This significant cytotoxicity can potentially cause a negative effect on the intrinsic functions and fates of MSCs. These results are consistent with a previous study that demonstrated the safety of Ac_4_ManNAz in stem cells [29]. Over 20 μM Ac_4_ManNAz treatment led to the inhibition of the functional properties of stem cells, such as proliferation rate, viability, rate of endocytosis, and genes related to cell adhesion. However, those effects by Ac_4_ManNAz treatment were not observed in MSCs when they were treated with 20 μM Ac_4_ManNAz. Therefore, we further optimized the appropriate treatment time of Ac_4_ManNAz via cellular fluorescence imaging of MSCs treated with 20 μM Ac_4_ManNAz; as described above, BCN-Cy5.5 was subsequently incubated with MSCs after Ac_4_ManNAz treatment to visualize the N_3_ on the cell surface (Figure 2d). The result indicates that N_3_ generation on the MSC surface was gradually increased in an incubation time-dependent manner, but the amount of N_3_ groups generated on the cell surface were nearly similar in MSCs after 48 or 72 h of Ac_4_ManNAz treatment. In addition, the amounts of BCN-Cy5.5 conjugated with N_3_ on the cell surface were significantly larger than the natural uptake of BCN-Cy5.5 that was confirmed in the MSCs without Ac_4_ManNAz treatment (Appendix A). From these results, we can expect that drug loading into the MSCs via metabolic glycoengineering and copper-free click chemistry could be considerably higher than conventional intracellular loading methods. Taken together, these results clearly demonstrate that treatment with 20 μM Ac_4_ManNAz for 48 h is the optimal condition to generate high amounts of N_3_ groups on the MSC surface without affecting their intrinsic functions and fates.

### 2.3. Preparation of AuNR-Incorporated MSCs (AuNR@MSCs) in Stem Cell Cultured System

Next, we prepared AuNR-incorporated MSCs (AuNR@MSCs) by the incubation of 20 μM Ac_4_ManNAz-treated MSCs with AuNR@BCN. To efficiently monitor the AuNR incorporation in the stem cells via fluorescence imaging, AuNR@BCN modified with NHS-Cy5.5 (Cy5.5-AuNR@BCN) was used, wherein the Cy5.5-AuNR@BCN has BCN and Cy5.5 groups with a ratio of 9:1. First, the different concentrations (0–200 μg/mL) of Cy5.5-AuNR@BCN were treated with the Ac_4_ManNAz-treated MSCs (Man^+^). The fluorescence signals (red color) of the AuNRs were clearly observed on the cell surface, and they became gradually stronger in a dose-dependent manner; however, those fluorescence signals in the AuNR@MSCs were similar after 200 or 400 μg/mL of AuNR@BCN treatment (Figure 3a). Notably, the amount of AuNRs incorporated into the cells was significantly higher in MSCs treated with Ac_4_ManNAz compared to naive MSCs (Man^−^). These results clearly indicate that metabolic glycoengineering-based nanoparticle incorporation allows a higher loading capacity in the stem cells than conventional intracellular loading methods. The AuNR incorporation efficiency was also evaluated at the different incubation times after the treatment of MSCs with 200 μg/mL AuNR@BCN (Figure 3b). The cellular fluorescence imaging results showed that the amount of AuNRs incorporated into the MSCs was nearly similar after 6 h and 12 h of AuNR@BCN treatment. The successful incorporation of AuNRs into the MSCs was further confirmed via cryogenic electron microscopy, which clearly shows nano-sized rod morphology on the cell surface after 6 h of 200 μg/mL AuNR@BCN treatment (Figure 3c).

Hence, the photothermal performance of AuNR@MSCs, which were prepared by the treatment of 200 μg/mL AuNRs for 6 h with MSCs pre-treated with 20 μM Ac4ManNAz for 48 h, was evaluated under light irradiation (808 nm, 1.0 W/cm^2^). The photothermal images clearly showed a potent heat generation efficiency of AuNR@MSCs (Man^+^/AuNR@BCN), wherein the local temperature in tubes was significantly increased up to 52 °C, along with the light irradiation time being longer (Figure 3d and Appendix A). Importantly, an increase in local temperature by AuNR@MSCs was significantly higher compared to MSCs incorporating AuNRs via conventional intracellular loading methods (Man^−^/AuNR@BCN; 39 °C) after 6 min of light irradiation with a power of 1.0 W/cm^2^. Taken together, the AuNR@MSCs prepared by the incorporation of AuNRs in the stem cells through metabolic glycoengineering and copper-free click chemistry reaction show considerable heat generation efficiency by allowing a high loading capacity of photothermal agents.

### 2.4. Deep Tumor Penetration of AuNR@MSCs in Colon Tumor Models

The deep tumor penetration of AuNR@MSCs was assessed in colon tumor models that were prepared by the subcutaneous inoculation of 1 × 10^7^ HT29 cells. When the tumor volumes were approximately 200 mm^3^, AuNR@MSCs or AuNR@BCN with an equivalent concentration of 5 mg/kg of AuNR were intravenously injected into mice.

Importantly, the tumor accumulation of AuNR@MSCs was significantly higher than that of AuNR@BCN, as confirmed by in vivo NIRF imaging of colon tumor-bearing mice (Figure 4a). The AuNR@MSCs in the tumor tissues was sustainably retained after 5 days of injection, whereas AuNR@BCN passively accumulated within the tumor tissues in the relatively lower levels and was rapidly removed from the tumors from 1 day of injection. These results are attributed to the natural homing effect by tumor tropism of stem cells. Quantitatively, the amount of AuNR@MSCs in the tumor tissues was 1.5–1.71-fold higher than AuNR@BCN on day 5 of injection (Figure 4b). Ex vivo NIRF images further showed 1.47–1.66-fold higher tumor accumulation of AuNR@MSCs than that of AuNR@BCN after 5 days of injection (Figure 4c). Histological analyses were additionally performed to confirm the deep tumor penetration of AuNR@MSCs. Interestingly, a strong Cy5.5 fluorescence (red color) of AuNR@MSCs was observed deep inside the tumor tissues along the CD31-positive blood vessels (green color) on day 5 of treatment (Figure 4d). Notably, AuNR@MSCs showed considerable accumulation in the whole region of the tumors compared with AuNR@BCN, indicating the efficient tumor accumulation of AuNRs by the MSC-mediated delivery strategy. More importantly, the Cy5.5 fluorescence (red color) of AuNR@MSCs was clearly observed in the central core region of the tumor tissues. These results demonstrate an effective deep tumor delivery of AuNRs by MSCs. As a control, only a little AuNR@BCN was observed in the tumor tissues owing to the limited targeting efficiency of nanoparticles by TME. These observations clearly demonstrate that AuNR@MSCs efficiently accumulate deep inside the tumor tissues via the intrinsic homing nature to tumors of stem cells [18]. From these results, we can also expect that AuNR@MSCs would promote a potent PTT under light irradiation owing to their considerable accumulation in whole tumor areas.

### 2.5. Therapeutic Efficacy of PTT by AuNR@MSCs in Colon Tumor Models

The therapeutic efficacy of PTT by AuNR@MSCs under light irradiation was assessed in HT-29 colon tumor models. First, photothermal efficiency to generate heat under light irradiation was evaluated in mice (Figure 5a). Briefly, the mice were randomly divided into four groups of (i) saline, (ii) light irradiation only (Laser only), (iii) AuNR@BCN with light irradiation (AuNR+L), and (iv) AuNR@MSCs with light irradiation (AuNR@MSC+L). When the tumor volumes were approximately 100 mm^3^, AuNR@MSCs or AuNR@BCN with equivalent concentration of 5 mg/kg of AuNR were intravenously injected into the mice. In addition, tumor tissues in the Laser only, AuNR+L, and AuNR@MSC+L groups were locally irradiated by light with a power of 1.0 W/cm^2^ for 5 min; light irradiation was performed after 3 days of AuNR@MSCs or AuNR@BCN treatment. The photothermal images of the mice showed significant hyperthermia in the tumor tissues in the AuNR@MSC+L group (47.4 °C) compared to the saline (35.4 °C), Laser only (40.7 °C), and AuNR@BCN+L (43.4 °C) groups. The high photothermal efficiency in localized tumor tissues is attributable to the high loading capacity of AuNR@MSC and their deep tumor penetration, resulting in considerable accumulation. Next, the therapeutic efficacy was assessed by monitoring tumor growth after treatment with the same protocol as described above (Figure 5b). As expected, the mice in the AuNR@MSC+L group (396.06 ± 10.42 mm^3^) showed significantly delayed tumor growth compared to those treated with saline (1758.2 ± 380.12 mm^3^), Laser only (1190.91 ± 290.75 mm^3^), and AuNR@BCN+L (857.08 ± 309.81 mm^3^) on day 18 of treatment. The limited therapeutic efficacy of AuNR@BCN+L is due to the low tumor accumulation by highly dynamic and complex ECM that hinder deep tumor penetration of nanoparticles. With the intrinsic biocompatible characteristics of AuNPs, the body weights of mice in the AuNR@MSC+L or AuNR@BCN+L groups showed no significant changes compared to those in the saline group (Figure 5c). As a result, the median survival of mice in the saline, Laser only, and AuNR@BCN+L groups was determined to be 20, 24, and 28 days, respectively; wherein the mice were dead owing to the tumor progression (Figure 5d). In contrast, mice treated with AuNR@MSC+L all survived over 30 days and had significantly inhibited tumor growth. Consequentially, AuNR@MSC can promote intense hyperthermia in the tumor tissues under light irradiation via the MSC-mediated deep tumor delivery of AuNPs, leading to a potent PTT.

## 3. Conclusions

In this study, we proposed MSC-mediated deep tumor delivery of AuNRs to trigger a potent PTT in colon tumors. First, AuNRs modified with bio-orthogonal click molecules (AuNR@BCN) were prepared, and their size distribution, morphology, photothermal efficiency, and biocompatibility were assessed in vitro. Then, AuNR@BCN was incorporated into the MSC via metabolic glycoengineering and copper-free click chemistry reaction. From these experiments, an appropriate condition to incorporate AuNR in the stem cells was carefully optimized. Importantly, the resulting AuNR@MSC showed high loading capacity compared to the conventional intracellular loading method, resulting in enhanced photothermal efficiency under light irradiation. Notably, the AuNR@MSC efficiently accumulated deep inside the tumor tissues owing to the tumor homing effect by the natural tumor tropism of stem cells when they were intravenously injected into the colon tumor models. As a result, the AuNR@MSC significantly inhibited colon tumor growth under local light irradiation. Overall, these findings suggested that MSC-mediated deep tumor delivery of AuNPs provide a new route for effective PTT. As such, this study introduced a novel technology for efficient delivery of nanoparticles using stem cells, which is potentially applicable for the treatment of a broad spectrum of diseases that require high targeting and deep tissue penetration.

## Data Availability

All relevant data are available with the article and its Appendix A, or are available from the corresponding authors upon reasonable request.

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
