# Peer review of "Mesenchymal Stem Cell-Mediated Deep Tumor Delivery of Gold Nanorod for Photothermal Therapy"

_nanomaterials, 2022, doi:10.3390/nano12193410_

Round 1

Reviewer 1 Report

The authors proposed MSC-mediated AuNPs deep delivery system through biorthogonal click molecules modification on AuNPs and further incorporated on MSCs via metabolic glycoengineering and copper-free click chemistry reaction for application of PTT. This work is interesting and the findings are useful for future application. It can be considered for publication after the following minor problems to be solved.

1.     On Fig. 5b, no error bars of AuNP@MSC+L, please confirm it.

2.     For therapy method of AuNP@MSC+L, how to control the concentration of AuNP to 5mg/kg? In other words, how to determine the amount of AuNPs in MSCs?

3.     CT26 cells were mentioned to construct tumor-bearing mice on Page 6, why it changes to HT29 cells on Page 12? Please confirm it and check the whole article.

4.     The mechanism of AuNPs deep delivery has not been clarified according to the in vitro and in vivo experiments.

Author Response

The authors appreciated all reviewers’ valuable comments. After a careful examination of those comments, we have prepared the revised manuscript to reflect them. Our responses to the reviewer’s comments are summarized in the attached file.

Reviewer 2 Report

Interesting work.

Following are some of the queries that needs to be addressed.

1) If the viability of MSCs after incubation with Ac4ManNAz treatment for 48 h is cytotoxic at 50uM, the authors have used 20 M Ac4ManNAz, it is bound to be toxic. I see all quantities of Ac4ManNAz in M. Kindly confirm.

2) Formation of N3 groups on the cell surface is specific to MSC or any other cells. 

3) What is the % of success for conjugation. AuNR stabilized with CTAB are cytotoxic. What if residual AuNR stabilized with CTAB remains in the mixture. Have the authors quantified the final yield after conjugation.

4) Line 158: Kindly elaborate how N3 generation in the MSCs be concluded after observing via confocal. Cy5 is hydrophobic fluorphore and may have a tendency to stain cell surfaces as such. A control of only dye for non-specific binding is needed.

5) Line 187 and 202 : Kindly confirm intranenously or intravenously? Also Line 240:  chagrge

6) Methodology is needed for processing the tumor tissues to obtain histology sections

Author Response

Manuscript ID: nanomaterials-1949719

Title: Mesenchymal stem cell-mediated deep tumor delivery of gold nanorod for photothermal therapy

The authors appreciated all reviewers’ valuable comments. After a careful examination of those comments, we have prepared the revised manuscript to reflect them. Our responses to the reviewer’s comments are summarized in the attached file.

Reviewer 3 Report

This manuscript is a study on using a MSC-mediated deep tumour delivery of gold nanorod (AuNR) for photothermal therapy (PTT). Experimental verification was carried out using colon tumour-bearing mice with intravenous injection. This work is topical and timely. I only have the following minor comments:

1.       Introduction, L40-43: When mentioning the applications of gold NP in cancer therapy (e.g. PTT), please consider to quote some more updated references such as Siddique et al (Nanomaterials 2020;10:1700) and Siddique et al (Nanomaterials 2022;12:2826)

2.       Introduction, L43-45: When mentioning the NP variables for effective PTT delivery, please consider to use updated reference such as Moore et al (Nano Ex 2021;2:022001)

3.       Scheme 1a and 1b: These two figures are joined together. The authors may consider only using one subfigure or really separate them into two subfigures.

4.       Preparation and characterization of AuNR@MSCs: For the synthesis of the AuNR@MSCs, it is good to have a schematic diagram showing how to link the AuNR to the MSC.

5.       Statistics, L209-214: Please state the software to perform the statistical tests.

6.       Figure 1a: It is good to have the scale bar in all four subfigures. Figure 1g: Please provide error bars for the data points.

7.       Figure 5a: For the second subfigure, “Lasor only” should read “Laser only”.

8.       Conclusions: The authors should mention the future prospectives of this study, and what will be the next step of this study.

Author Response

(The authors gave the same response as above.)

Round 2

Reviewer 3 Report

I am satisfied with the corrections and modifications from the authors based on my comments. I am also satisfied with the modified figures and their explanations to my concerns. The quality and presentation of the manuscript are improved.

Author Response

The authors greatly appreciated to valuable reviewer's comments.
